# A new method for the reproducible development of aptamers (Neomers)

Cathal Meehan[1], Erika L. Hamilton[2], Chloé G. Mansour[2], Soizic Lecocq[1], Cole J. Drake[2], Yisi An[2], Emily Rodrigues[2], Gregory Penner[1,2]*

1 NeoVentures Biotechnology Europe SAS, Villejuif Bio Park, Villejuif, France, 2 NeoVentures Biotechnology Inc., London, Canada

* gpenner@neoventures.ca

## Abstract

The development of aptamers has been almost exclusively performed based on the SELEX method since their inception. While this method represents a powerful means of harnessing the in vitro evolution of sequences that bind to a given target, there are significant constraints in the design. The most significant constraint has been the reliance on counter selection on off-targets to drive specificity. Counter selection has not been as effective at driving aptamer specificity as the presence of immune tolerance, the capacity of the immune system to remove antibodies that bind to host targets, is for antibody development. This deficiency has constrained the commercial efficacy of aptamers to date. These limitations have been addressed with our design of a novel platform for aptamer identification. This new design is based on what we refer to as a Neomer library with sixteen random nucleotides interspersed with fixed sequences. The fixed sequences are designed to minimize the potential for hybridization, such that secondary structure is driven by the random nucleotides. The use of sixteen random nucleotides reduces the possible library sequences to $4.29 \times 10^9$. This enables the application of the same sequences to either the same target or different targets while maintaining a high level of structural diversity. In effect, this introduces the capacity for reproducibility in aptamer selection and an in-silico approach to replicating immune tolerance. We provide here an overview of the new method and a description of the performance of aptamers selected for interleukin 6 developed using this approach.

## Introduction

The invention of aptamers by four different groups between 1989–1990 [1–4] introduced the principle of Systematic Evolution of Ligands by EXponential enrichment (SELEX) for the enrichment of DNA or RNA oligonucleotide sequences from random libraries for specific targets. Since then, there has been continued improvement of the SELEX process to improve aptamer specificity and affinity. In 1992 Ellington et al. introduced negative SELEX with counter selection against the solid support used to immobilize the target [5] and in 1994, Jenison et al. added counter selection against similar targets to enhance specificity [6]. SELEX methodology has also been applied in different platforms including capillary electrophoresis (CE) SELEX [7,8], which increases the stringency of selection; microfluidic SELEX [9,10],

**Data availability statement:** All relevant data are within the manuscript and its Supporting information files.

**Funding:** The author(s) received no specific funding for this work.

**Competing interests:** Conception and design of library: G.P. Optimization of library selection and NGS processing approaches, G.P., E.L.H., and S.L. Acquisition, analysis or interpretation of data: G.P., C.M., C.G.M., C.J.D. and Y.A. Writing and/or revisions of the manuscript: G.P., C.M., and C.G.M. All authors have approved the submitted version of the manuscript; agree to be personally accountable for their own contribution; and commit to any action that ensures or assesses integrity of any part of the work. C.M. and S.L are employees of NeoVentures Biotechnology Europe SAS, a privately held company that applies aptamer libraries to identify bio-markers. E.L.H., C.G.M., C.J.D., and Y.A are employees are of NeoVentures Biotechnology Inc., a privately held company that develops aptamers for defined targets. G.P. is an owner of NeoVentures Biotechnology Inc. which in turn wholly owns NeoVentures Biotechnology Europe SAS. This does not alter our adherence to PLOS ONE policies on sharing data and materials.

which reduces the amount of target required for selections; and Cell SELEX [11], which allows for selection of targets in their native environment and agnostic selection against cell types. These adaptations of SELEX provide many advantages, but they are still constrained by their core reliance on the SELEX concept.

The SELEX procedure consistently has a library design featuring a contiguous random region flanked by two polymerase chain reaction (PCR) primer recognition sequences. The size of the random region varies from 25 to 80 nucleotides (nt) and depends on different factors, such as the size of the target [12]. Research indicates that the length of the random sequence directly influences the structural diversity of the SELEX library, with smaller random regions yielding less structural diversity [13]. A minimum number of homologous nucleotides is necessary to form stable hybridizations, and the hybridization of at least three nucleotides between different regions of the aptamer are required for the formation of hair-pin loops [14].

To address basic limitations of the SELEX sampling, we can consider a library of 40 random nucleotides, a common contiguous random nucleotide length from commercial and academic settings [15]. This number of random nucleotides has a possible sequence space of $1.2 \times 10^{24}$ ($4 \times 10^{40}$). It is impractical to work with this many sequences in a selection process, selections generally start with an initial naïve library containing $1 \times 10^{15}$ random sequences. This subset represents a proportion of $8.27 \times 10^{-10}$ of the total sequence solution space. Such a small subset of the total sequence solution space creates two constraints. First, there is an average of one copy for each unique sequence in the initial naïve library before the first round of selection. This means that there is a possibility that sequences that would show binding to the target aren't identified due to stochastic variation in the binding solution space. This also means that multiple iterative rounds of selection are needed for sequences to sufficiently enrich for adequate characterization. Prior to each subsequent selection round, it is essential to amplify the selected library which introduces additional technical variation due to PCR bias [16] into the selection process. Secondly, the small subset of the total sequence solution space means that selection is not reproducible. It is not possible to start selection with the same subset of initial naïve sequences.

A key biological difference between antibody development and aptamer development is the presence of immune tolerance within the antibody development system. Immune tolerance is the capacity of the immune response system to remove antibodies that bind to pre-existing host epitopes, thus preventing the immune system from damaging the host [17]. The in vitro nature of aptamer development means that this step is not present. Moreover, the lack of reproducibility implicit in SELEX means that it has not been possible to create an in-silico approach that would reproduce the effects of immune tolerance.

Counter selection was introduced to the SELEX process to address this limitation. Counter selection works well at removing sequences that bind strongly to the counter selection target, but effectiveness decreases as cross reactivity decreases. Furthermore, the weaker a given aptamer sequence cross-reacts to a counter selection target the lower the proportion of that aptamer that would be removed by the counter selection process [18].

This is a fundamental problem given the magnitude of differences in the abundance of proteins in biological matrices and target proteins. For example, human serum albumin (HSA) is present in human blood at a concentration range of 522–746 μM [19]. If a target molecule was present at a concentration of 600 pM, then there would need to be a specificity of a million-fold towards the target over HSA, or the aptamer would be saturated by the weak binding to HSA over the aptamer. Counter selection against aptamer sequences that bind to HSA with an affinity that is 10,000 to 100,000-fold less strong than their affinity to the target molecule will not be effective in reducing the enrichment of such sequences in SELEX selection.

The development of highly specific aptamers for use in complex biological matrices presents significant challenges, particularly when targeting low-abundance proteins such as interleukins. Recent analyses of aptamer literature for interleukin targets shows very few aptamers developed that include specificity data or evaluate their performance in complex biological mixtures [20]. This lack of rigorous testing is particularly problematic for targets like IL6, where the specificity requirements are exceptionally stringent due to their low physiological concentrations.

Traditional counter-selection methods in SELEX often prove inadequate for eliminating aptamers with even modest affinity for abundant off-target proteins. This phenomenon likely contributes to the limited success of aptamers in commercial therapeutics and diagnostics to date. We postulate that the binding of aptamers to abundant proteins in biological matrices has been the primary reason for this lack of commercial success.

We have changed aptamer selection through a fundamental redesign which we have termed the Neomer library. This library design comprises sixteen random nucleotides interspersed with fixed sequences, thus reducing the number of possible sequences to $4.29 \times 10^9$. This enables the application of the same set of sequences in selection against different targets or in replication against the same target. This allows for the identification of aptamers that bind with the highest affinity and specificity possible to the target. In this paper we describe the successful use of a Neomer library for the development of aptamers for interleukin 6 (IL6). We also provide a process for the in-silico evaluation of the performance of the selected IL6 sequences in a selection against HSA.

The SELEX method has been the only approach for aptamer development since its inception. In summary, SELEX has the following limitations:

1. Lack of reproducibility: Due to sampling only a small subset of possible sequences in a synthesized library ($1 \times 10^{15}$ within $1.2 \times 10^{24}$ possible sequences)

2. Lack of statistical robustness: Only one copy per sequence allows potentially effective sequences to be lost to stochastic processes due to the low probability of binding between the target and any one sequence

3. Limited secondary structure: Contiguous random region implicitly limits secondary structure formation as random nucleotides within three nucleotides of each other are not able to hybridize with each other. Additionally, the requirement for longer primer recognition sites, constrains the structural diversity per position in these regions to overcome potential hybridisation within the contiguous random region. Longer regions not necessary for binding can impair the effective function of aptamer motifs responsible for aptamer-target binding.

4. Multiple rounds of selection and amplification required: There is single strand recovery bias in each selection cycle to purify the sense strand from the antisense strand. Sequences with higher G/C content are implicitly more difficult to separate and thus are biased against.

5. Ineffective counter selection for specificity: The non reproducible sequence set prevents in silico screening for specificity.

To address these limitations, we developed the Neomer library approach, which utilizes a template with interspersed fixed and random regions. The design process aimed to optimize structural diversity and statistical robustness. We selected 16 random nucleotides, resulting in $4.29 \times 10^9$ possible sequences. This number allows for comprehensive coverage in a single NGS run, enables multiple copies of each sequence in the initial library, and

facilitates effective single-round selection. The fixed regions were designed to minimize self-hybridization while incorporating necessary functional elements such as primer recognition sequence sites and a restriction enzyme site for module separation:

1. Reproducible selection: The closed sequence set of 16 random nucleotides that results in $4.29 \times 10^9$ (4^16) enables, for the first time, reproducible selection of aptamers

2. Statistical robustness: This ensures approximately 1000 copies of each possible sequence is applied in a selection, ensuring sufficient probabilistic interactions between targets and aptamers for robust statistical evaluation

3. Structural diversity: The interspersed design of fixed sequence and random nucleotides maximises the structural diversity introduced by random nucleotides by minimizing self-hybridisation between fixed regions and allowing sufficient structural flexibility by ensuring a GC content between 40–60% and high minimum free energy (MFE).

4. Single-round selection: The design allows for effective reproducible single-round selection, reducing PCR bias and unique sequence solution space effects by ensuring the same pool of initial sequences are applied to the target each time

5. In silico specificity screening: The closed sequence set enables in silico screening for specificity by comparing how every possible sequence in the library performs on counter targets, mimicking immune tolerance

This Neomer design addresses the limitations of SELEX while providing a reproducible, specific and statistically robust approach to aptamer selection.

## Results

The Neomer library was designed to enable single-round reproducible selection by reducing the number of random nucleotides, allowing for comprehensive evaluation of all sequences with appropriate copy numbers for robust statistical analysis. Sixteen random nucleotides were chosen, generating a possible set of $4.29 \times 10^9$ sequences. The Neomer selection process starts with a pool of $4.29 \times 10^{12}$ sequences, resulting in an average representation of 1,000 copies for each of the possible $4.29 \times 10^9$ sequences (Fig 1A). This initial redundancy allows for the assessment of selection effects on each sequence after a single round of selection, reducing the impact of PCR bias. It also ensures probabilistic collisions between aptamers and targets and enables statistically effective sequence evaluation in the subsequent NGS analysis.

Reproducible single round selection allows the assessment of the same initial set of sequences across multiple samples. This allows us to evaluate the frequency (read count normalised for total read count of sequences in a sample) of all $4.29 \times 10^9$ possible and evaluate aptamers that bind to the target versus a naïve control library (Fig 1B). However, a single NGS lane returns approximately $1 \times 10^9$ reads, which is less than the total number of possible sequences in our library. To adequately utilise NGS and address this constraint to carry out a statistically robust analysis, we developed a module driven approach. We incorporated a restriction digest site in the middle of the template to split it into two parts post selection (module A and B), each with eight random nucleotides or 65,536 possible sequences, and evaluated performance on these modules with NGS analysis.

To facilitate effective neomer selection and maximize structural diversity within the constraints of 16 random nucleotides the following methodology was implemented:

1. Primer recognition sequences were designed considering 40-60% G/C content and ensuring non-hybridization.

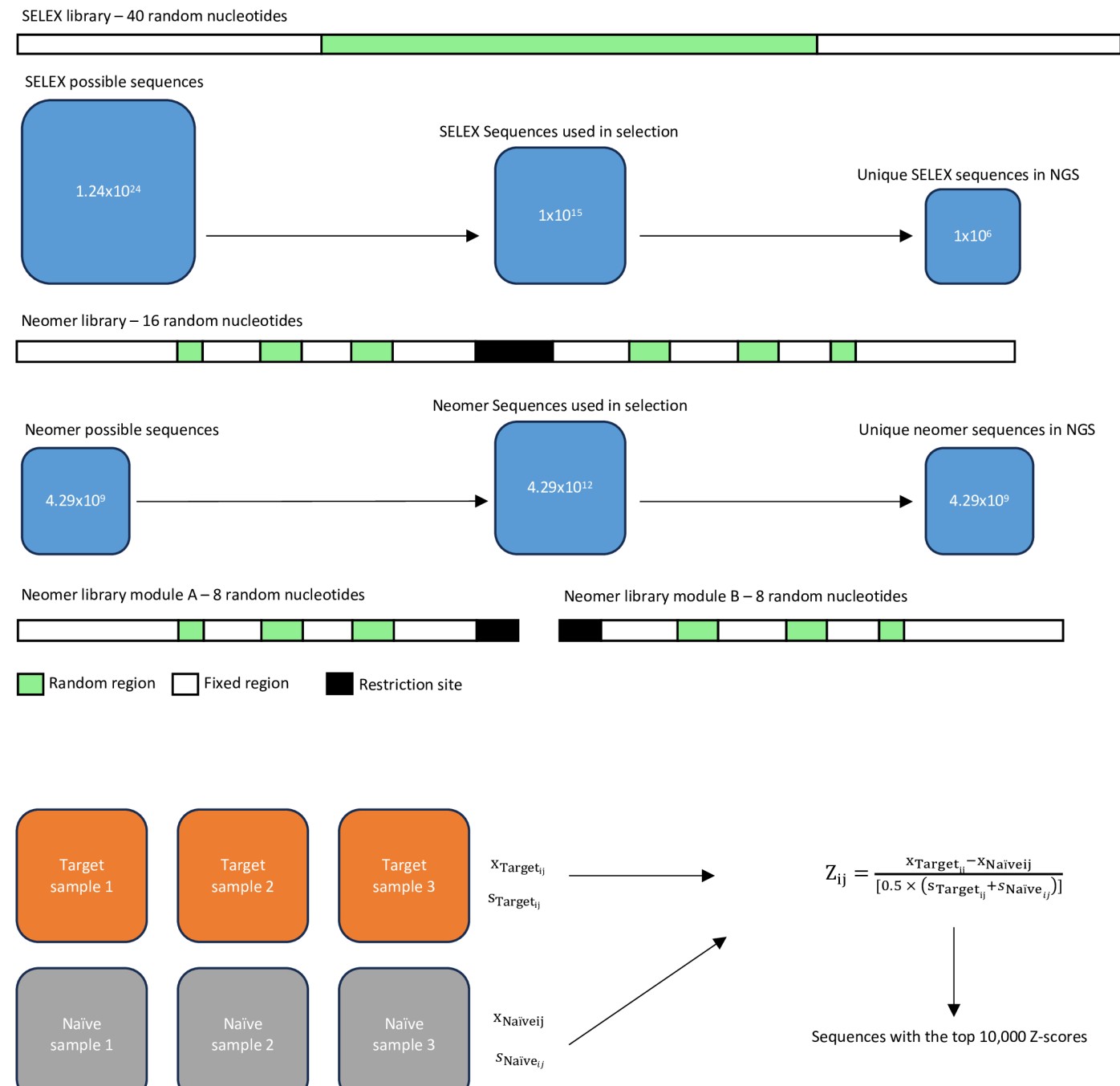

**Fig 1. Visualization of the differences between SELEX and Neomer selection.** A) A schematic showing the templates of SELEX and Neomer libraries respectively. A legend detailing the regions is provided at the bottom. Possible sequences, sequences used in selection and the number of unique sequences is shown in proportion below each template. B) A schematic showing how the z-score metric is evaluated in the Neomer approach. Target replicates represent samples that have undergone selection against a target or condition or interest, while naïve replicates represent samples that haven't gone through selection but could also represent any counter target condition.

2. A restriction site was incorporated in the middle of the library for module separation, with the surrounding fixed region designed for dual functionality as both a restriction site and a primer recognition site post-restriction.

3. Internal sequences for primer recognition were designed following standard primer rules.

4. Random nucleotides were arranged in a symmetrical manner (2:3:3) with consistent fixed sequence nucleotide spacing between such regions. This approach of using blocks of random sequences (2 or 3 nucleotides) instead of single nucleotides was selected to drive higher levels of structural diversity, as hybridisation is more likely to occur contiguously for larger secondary structures.

5. Fixed sequences were designed using a random identity generator, with nucleotide substitutions made as necessary to avoid internal hybridization and keep minimum free energy (MFE) sufficiently high, preventing energy barriers to secondary structure formation.

This design principle optimizes the structural diversity of the library by minimizing inherent secondary structure in the fixed regions. The absence of pre-existing hybridizations between fixed region nucleotides reduces energetic barriers that could impede the ability of sequences to adopt alternative conformations necessary for target binding. This approach is supported by previous research indicating that libraries with greater structural flexibility demonstrate improved aptamer selection outcomes compared to highly structured libraries, suggesting that an initially unstructured state facilitates more diverse target interactions [21].

Consequently, the secondary structure in this library design is primarily driven by the identity of the random nucleotides, maximizing the potential for unique structures within the total library diversity. This strategy aims to enhance the likelihood of identifying high-affinity aptamers by allowing the random regions to explore a wider range of conformations during the selection process. Assessment of secondary structure in the Neomer library design was performed with Vienna RNAfold [22].

To assess the impact of reduced random nucleotides on structural diversity in the Neomer library, a comparative analysis with a SELEX library with 40 random nucleotides and Somalogic primer recognition sequences was conducted [15]. While a reduction to sixteen contiguous random nucleotides in a SELEX library would significantly constrain structural diversity, the Neomer library design mitigates this limitation. The SELEX library's contiguous random region implicitly constrains secondary structure formation due to the limited folding capacity of adjacent nucleotides. For example, a sequence 'GGGCCC' can be represented as '......' in dot-bracket notation, where '.' denotes an unhybridized nucleotide. Inserting nucleotides, as in 'GGGNNNCCC', transforms the structure to '(((...)))', where '(' and ')' represent the 5' and 3' sides of a hybridized region, respectively. This demonstrates how the contiguous nature of the first sequence constrains secondary structure formation, a constraint that is mitigated by interspersing random nucleotide positions between potentially hybridizing bases.

A direct assessment of structural diversity between the Neomer library and a SELEX library was performed to evaluate the distribution of secondary structures across both templates. We predicted and annotated the secondary structure of 1000 randomly generated sequences using for both templates using Vienna RNAfold and bpRNA and visualised them in a heatmap to investigate if there were any structural or positional biases in representation in either library [23,24] (Fig 2A and 2C). We can see comparable levels of diversity between both library designs. Despite the reduction in random nucleotides, the Neomer library design showed higher structural diversity in the primer recognition regions. It should be noted that dangling ends, which are more prevalent in the SELEX library design and are characterised as a lack of secondary structure, are less likely to form complex functional motifs compared to other

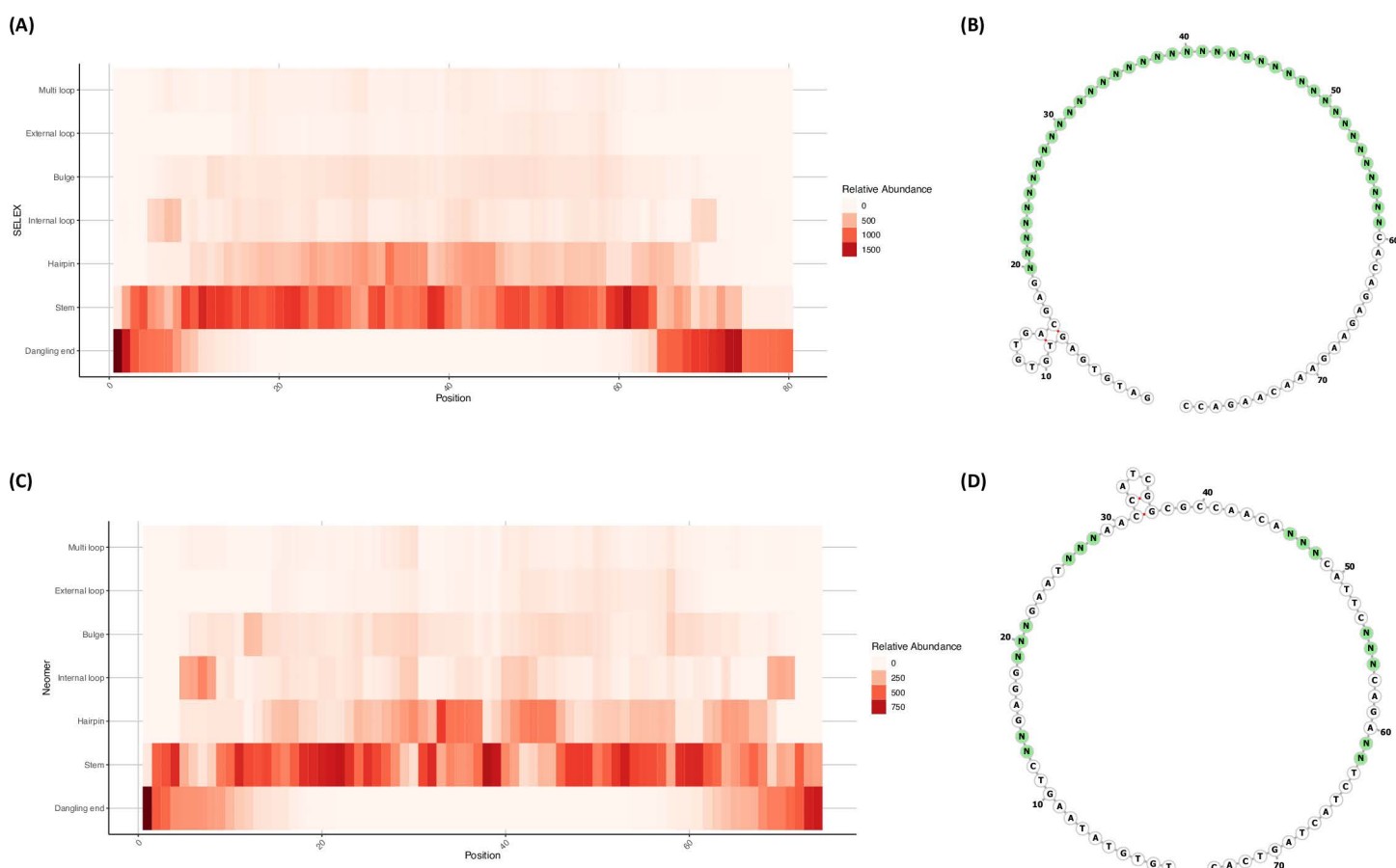

**Fig 2. Comparison of the predicted secondary structures for SELEX and Neomer libraries.** A) Heatmap plot of the distribution of secondary structure motifs across 1000 randomly generated SELEX template sequences. Relative abundance is detailed in the legend on the right. B) FORNA RNAfold plotting of the secondary structure of the SELEX template sequence. Random nucleotides are shown in green. C) Heatmap plot of the distribution of secondary structure motifs across 1000 randomly generated Neomer template sequences. Relative abundance is detailed in the legend on the right. D) FORNA RNAfold plotting of the secondary structure of the Neomer template sequence. Random nucleotides are shown in green.

secondary structures such as hairpins or bulges, although they have been shown to contribute to stability in siRNA [24]. Both libraries showed comparable proportions of secondary structures and secondary structure length except for dangling ends and stems (S1 Fig). Secondary structure diversity by position was also evaluated in both templates, showing comparable levels of diversity, with Neomer showing a more punctuated pattern of diversity but higher in primer recognition regions (S2 Fig). Visualisation of random nucleotide templates with FORNA showed that the neomer template minimised pre-existing hybridisation between fixed regions (Fig 2) [25].

NGS analysis was performed separately on each module of the library, enabling sufficient NGS read coverage to predict the frequencies of each possible sequence per module (Fig 1B). The predicted frequency for each possible sequence within each module were then combined using an outer product operation (analogous to creating a Punnett square) to generate a predicted frequency for each of the $4.29 \times 10^9$ sequences in the full possible sequence space.

The neomer library design enables repeated application of the same initial naïve sequences to a single target, as well as application to multiple different targets. The selection was performed with the same library on both a target and on a negative control, with triplicate replicates (Fig 1B).

NGS analysis yielded an average of $16 \times 10^6$ reads for each module A and B sample (S1 Table). A read depth exceeding $5 \times 10^6$ in each module was deemed sufficient for sequence coverage and subsequent analysis. This read depth corresponds to an average of 76 copies per possible sequence. Sequence frequencies were complied across replicates to calculate mean frequency and standard deviation for each sequence.

A z-score metric was utilized to evaluate the performance of each of the $4.29 \times 10^9$ possible sequences in selection. The z-score was calculated as follows: (average frequency in target condition – average frequency in naïve or counter target condition) divided by the average of the standard deviation from both conditions. This calculation was performed for each sequence using the average predicted frequencies and standard deviation from both the positive target (IL6) replicates and the negative control (initial naïve library) replicates. The top 10,000 sequences identified to have the highest z-scores were retained for further analysis.

The distribution of fold values (IL6/naïve -1) versus the top 10,000 sequences based on z-score values were investigated. Sequences exhibiting both high fold and high z-score values were expected to perform optimally as candidate aptamers.

The use of the same $4.29 \times 10^9$ sequences enabled screening for the performance of these top 10,000 sequences on several counter targets, including HSA, the UltraLink resin used for HSA immobilisation, and the nickel resin used for IL6 protein immobilisation. The performance of these sequences on counter targets was evaluated by comparing the frequency predicted on the positive target to the frequency predicted on each counter-target as a fold value (Fig 3A).

The overall distribution of fold differences between the IL6 sequences and counter-targets (HSA and UltraLink) was similar, implying that sequences exhibiting high specificity against HSA also exhibited high specificity against the UltraLink resin (Fig 3). A high level of concordance between the fold values for the top 10,000 sequences in IL6 versus HSA and UltraLink was observed. This concordance was not apparent in comparison to the nickel resin used for IL6 immobilization, indicating that certain sequences from the IL6 selection were selected based on their capacity to bind to the nickel resin rather than IL6. Multiple filters were required to identify the top candidate sequences in terms of specificity against all targets. Four candidate sequences were selected based on high performance and specificity against this group and nickel resin (IL-6-9805, IL-6-4202, IL-6-6449 and IL-6-7326) (Table 1). Secondary structures for these sequences were predicted and visualised for high structural diversity amongst the candidates (Fig 4).

Prior to performing binding assays, candidate sequences IL-6-4202 and IL-6-7326 were truncated by removing a portion of the dangling ends and visualised with FORNA (Fig 4) [25]. The truncations in both cases effectively removed module B from the sequence. All the top fifteen sequences listed in Table 1 were observed to contain the same B module sequence. High structural diversity was observed across candidate aptamers despite the nucleotide consistencies of the random regions in module B (Table 2). These nucleotides specifically contributed to varying structural motifs within the aptamers such as internal and external loops and stems.

We injected varying concentrations of IL6 and HSA in flow over the candidate aptamers as well as a negative control. The resonance due to binding is the result of the total resonance measured on the candidate aptamers minus the resonance measured on the negative control (Fig 5) All four sequences exhibited strong binding affinity and high specificity to IL6 (Table 3). No binding was quantified when tested against HSA.

## Discussion

A novel approach to aptamer development has been demonstrated to effectively identify aptamers with sufficient binding affinities and specificity to be of interest for commercial application in diagnostics.

# IL–6 Aptamer Analysis

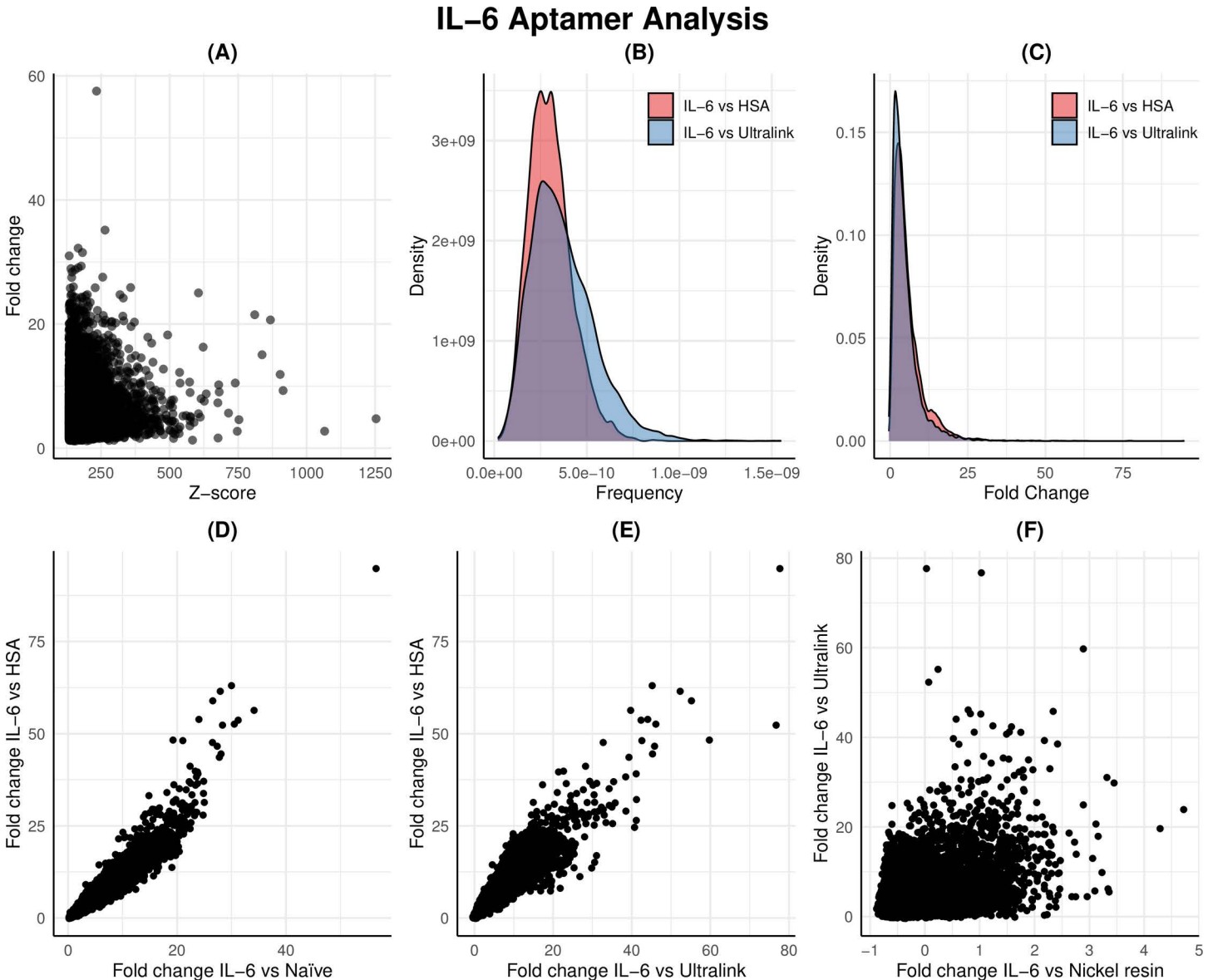

**Fig 3. Analysis of NGS results for IL-6 aptamer selection.** A) Scatterplot showing the relationship between Z-score and fold change in the top 10,000 sequences from the IL-6/Naïve selection. B) Density plot comparing the frequency of the top 10,000 IL-6/Naïve selection sequences in IL-6/HSA and IL-6/Ultralink selections. C) Density plot comparing the fold change of the top 10,000 IL-6/Naïve selection sequences in IL-6/HSA and IL-6/Ultralink selections. D) Scatterplot comparing fold change values of the top 10,000 sequences between IL-6/Naïve and IL-6/HSA selections. E) Scatterplot comparing fold change values of the top 10,000 sequences in the IL-6/Ultralink and IL-6/HSA selections. F) Scatterplot comparing fold change values from the top 10,000 sequences in the IL-6/Nickel resin and IL-6/Ultralink selections.

The Neomer library introduces an innovative design that is composed of 16 random nucleotides interspersed with fixed sequences. The separation of the random nucleotides with fixed sequences that are designed to hybridize to each other enables a maintenance of structural diversity when compared to 40 nucleotide contiguous random regions in a SELEX library. The reduction in the number of possible sequences in the Neomer library enables reproducible selection of the same sequences. In addition, by reducing the sequence solution space, it is possible to initiate selection with multiple copies of each possible sequence, enabling a single round of selection. It is possible that 16 random nucleotides are not the optimal

**Table 1. IL 6 fold values for all contrasts.**

| Sequence ID | IL6/HSA FC | IL6/UL FC | IL6/Naïve FC | IL6/Nickel FC |
|---|---|---|---|---|
| 1112 | 94.79 | 77.68 | 56.55 | 0.03 |
| 9805 | 63.04 | 45.21 | 30 | 1.02 |
| 8230 | 61.5 | 52.32 | 27.94 | 0.07 |
| 794 | 58.92 | 55.19 | 26.56 | 0.24 |
| 698 | 56.33 | 39.73 | 34.15 | 0.52 |
| 22 | 53.88 | 44.07 | 24.02 | 0.57 |
| 4202 | 53.67 | 42.37 | 31.23 | 1.58 |
| 2975 | 52.6 | 46.13 | 30.52 | 0.79 |
| 3290 | 52.32 | 76.73 | 28.35 | 1.03 |
| 6449 | 48.27 | 59.75 | 19.26 | 2.89 |
| 9076 | 48.12 | 42.58 | 21.06 | 1.24 |
| 7598 | 47.61 | 32.74 | 26.51 | 1.79 |
| 7326 | 46.6 | 45.81 | 27.38 | 2.34 |
| 4191 | 44.51 | 45.29 | 28.08 | 0.83 |
| 5489 | 43.6 | 39.3 | 27.77 | 2.18 |

balance between structural diversity and statistically robust reproducible selections, we plan to address this question in the future. However, applying the Neomer method with 16 random nucleotides resulted in the successful identification of four aptamers with high selectivity and specificity for IL6.

The outer product calculation does not have the capacity to fully determine individual sequence frequencies. An artifact of this analysis method is the distribution of the enrichment of a given sequence among sequences that share a common module. The accuracy of sequence frequency prediction is affected by the magnitude of selection effect as a function of sequence enrichment. We have recognized that the addition of more replications of the naïve control library reduces sampling error and decreases statistical artifacts due to low standard deviation values across a limited number of replications.

By utilising a closed sequence set, the Neomer approach enables reproducible selection of aptamers for the first time. This reproducibility extends to both repeated selections against the same target and selections against different targets, providing a foundation for robust statistical analysis of aptamer enrichment that was previously impossible with SELEX. In contrast to antibody selection, which is constrained from inherent variability due to host-specific antibody repertoires and the stochastic nature of V(D)J recombination, the Neomer libraries offer a consistent starting point. This consistency allows for the accumulation of knowledge from multiple selection experiments from the same initial naïve repertoire.

The Neomer approach represents a pivotal advancement in aptamer development, addressing key limitations in statistical evaluation, specificity, and reproducibility. This innovation enables the identification of aptamers with enhanced sensitivity and specificity that bind to known biomarkers or agnostic targets specific for disease conditions, overcoming a central bottleneck in aptamer commercialization. The closed sequence set allows for comprehensive comparisons across different targets, facilitating the development of more complex objectives and in-silico models for aptamer-target interactions leading to improved predictive capacity in aptamer development.

The reproducibility inherent in the Neomer method introduces new opportunities for standardization in aptamer development, a crucial factor for their broader adoption in therapeutic

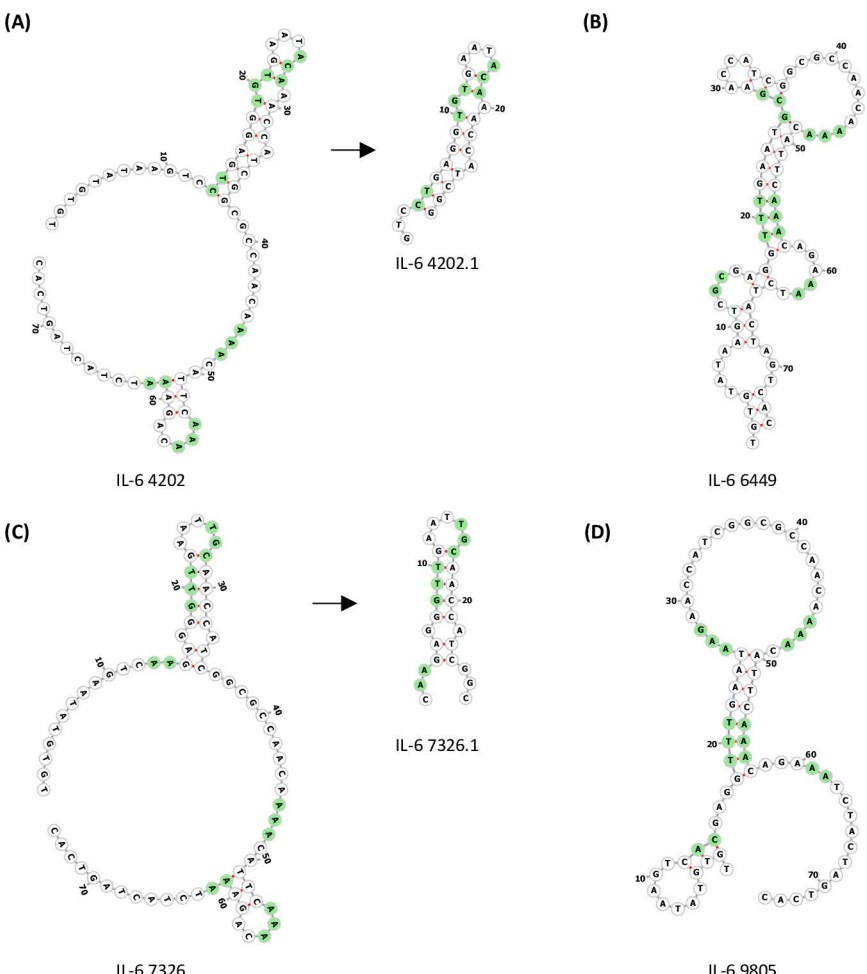

**Fig 4. Secondary structure visualisation of candidate aptamers.** A) FORNA visualisation of IL-6-4202 for plotting out secondary structure. Random nucleotide positions are labelled in green. A black box shows the truncated structure IL-6-4202.1. Secondary structure visualisation shows maintenance of the same structure from IL-6-4202. B) FORNA visualisation of IL-6-6449 for plotting out secondary structure. Random nucleotide positions are labelled in green. C) FORNA visualisation of IL-6-7326 for plotting out secondary structure. Random nucleotide positions are labelled in green. A black box shows the truncated structure IL-6-7326.1. Secondary structure visualisation shows maintenance of the same structure from IL-6-7326. D) FORNA visualisation of IL-6-9805 for plotting out secondary structure. Random nucleotide positions are labelled in green.

**Table 2. Identity of random module in IL-6 candidate sequences.**

|  | Module A | | | Module B | | |
| --- | --- | --- | --- | --- | --- | --- |
| Sequence ID | Region 1 | Region 2 | Region 3 | Region 1 | Region 2 | Region 3 |
| IL-6-4202 | CT | TGT | ACA | AAA | AAA | AA |
| IL-6-6449 | GC | TTT | GG | AAA | AAA | AA |
| IL-6-7326 | AA | GTT | TGC | AAA | AAA | AA |
| IL-6-9805 | AC | TTT | AAG | AAA | AAA | AA |

and diagnostic applications. In the context of our IL6 aptamer development, this approach has facilitated not only the selection of high-affinity aptamers but also the in-silico evaluation of their specificity against HSA.

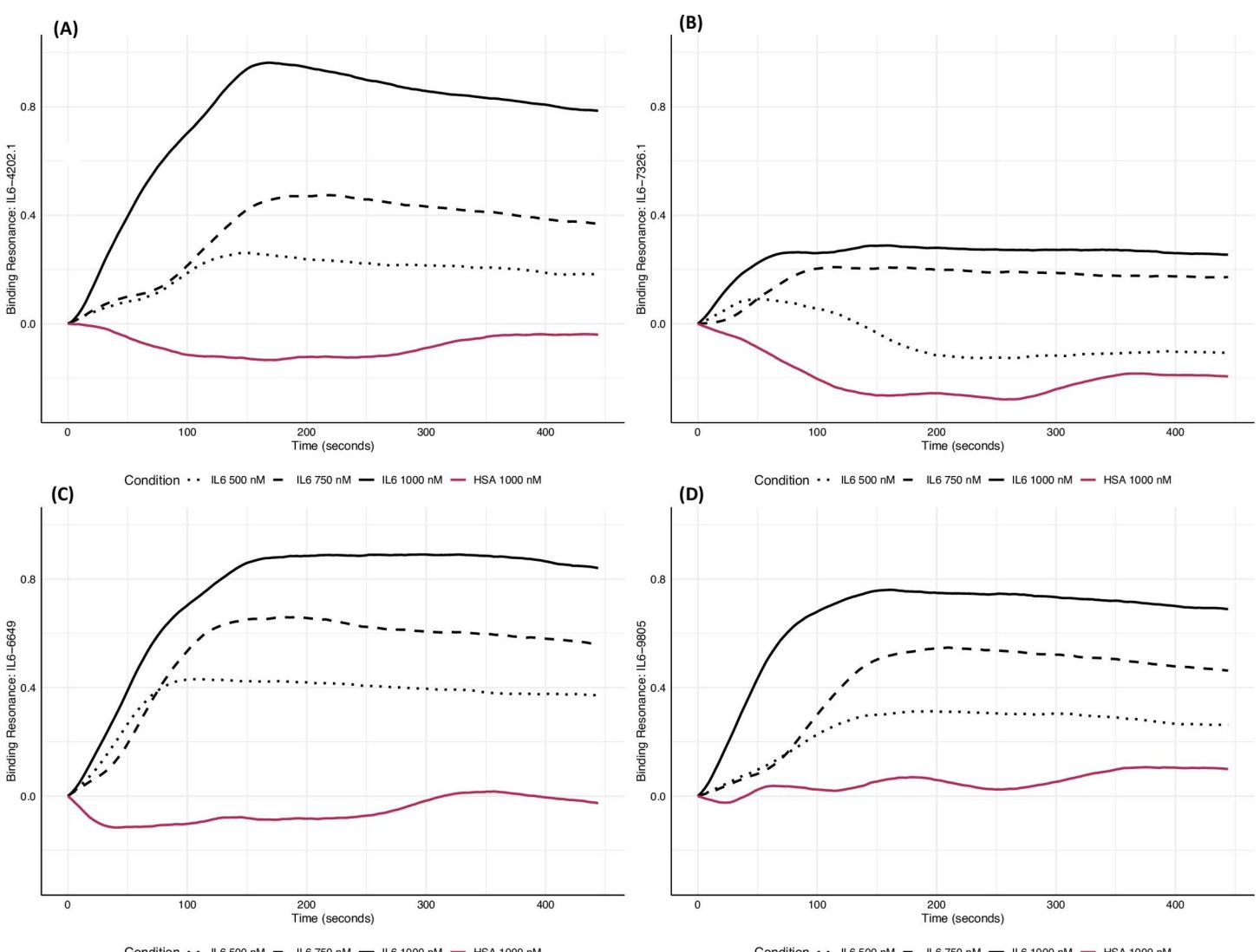

**Fig 5. SPRi results for IL6 aptamer sequences to IL6 and HSA.** A) Binding resonance of IL-6-4202.1 over time across a variety of concentrations of IL-6 detailed in the legend on the bottom. B) Binding resonance of IL-6-6649 over time across a variety of concentrations of IL-6 detailed in the legend on the bottom. C) Binding resonance of IL-6-7326.1 over time across a variety of concentrations of IL-6 detailed in the legend on the bottom. D) Binding resonance of IL-6-9805 over time across a variety of concentrations of IL-6 detailed in the legend on the bottom.

**Table 3. Binding coefficients for candidate aptamers versus IL 6 and HSA.**

| Sequence ID | | IL 6 | | | HSA |
|---|---|---|---|---|---|
| | $Kd$ ($s^{-1}$) | $ka$ ($M^{-1}s^{-1}$) | $kD$ (M) | $kD$ (M) |
| IL-6-4202.1 | 2.98E-03 | 4.07E$^+$04 | 7.32E-8 | NB |
| IL-6-6449 | 1.52E-03 | 4.73E$^+$04 | 3.21E-8 | NB |
| IL-6-7326.1 | 1.50E-03 | 5.56E$^+$04 | 2.70E-08 | NB |
| IL-6-9805 | 1.77E-03 | 4.37E$^+$04 | 4.05E-08 | NB |

NB, no binding measurable.

This innovation introduces the potential improvement of aptamer performance in diverse biological matrices, inciting a paradigm shift in the landscape of molecular recognition technologies.

## Materials and methods

### Library design

All DNA sequences including library, primers, and aptamers were synthesized by Integrated DNA Technologies with standard desalting. The Neomer library consists of a 73 nt long synthetic ssDNA with the following sequence:

5'-TGT GTA TAA GTC NNG AGG NNN GAA TNN NAA CCA TCG GCG CCA ACA NNN CAT TCN NNC AGA NNT CTA CTA GTC AC-3' where 'N' represents an equal probability of any nucleotide. This design resulted from a systematic process to optimize structural diversity and functionality. The library design process began with the creation of primer recognition sequences, considering 40–60% G/C content and ensuring non-hybridization. A KasI restriction site (CCATCGGCGCC) was incorporated in the middle of the template, serving a dual function as both a restriction site and a primer recognition site post-restriction. Internal sequences for primer recognition were designed following standard primer rules. Random nucleotides were arranged in a symmetrical manner (2:3:3) with consistent fixed sequence nucleotide spacing between such regions. This approach of using blocks of random sequences (2 or 3 nucleotides) instead of single nucleotides was selected to drive higher levels of structural diversity, as hybridisation is more likely to occur contiguously for larger secondary structures. Fixed sequences were designed using a random identity generator, with nucleotide substitutions made as necessary to avoid internal hybridization and keep minimum free energy (MFE) sufficiently high, preventing energy barriers to secondary structure formation. Fixed regions were selected iteratively using the RNAfold webserver at 22°C, 0.127M salt concentration, using DNA energy parameters and with avoiding isolated base pairs toggled to predict and minimize self-hybridisation, with the exception of the palindromic restriction digest site which separates the modules (23). The fixed regions were designed to have a relatively high MFE, which promotes structural flexibility and allows the identity of the random nucleotides to drive the formation of secondary structures. This approach increases the potential for diverse aptamer conformations. The overall GC content of the fixed regions was maintained between 40–60% to achieve a balance between stability and flexibility in the library structure. The arrangement of random nucleotides (NN or NNN) interspersed between fixed regions was designed to maximize potential structural diversity while maintaining library stability. The central CCATCGGCGCC sequence functions as a KasI restriction site, enabling module separation and comprehensive NGS analysis. This design allows each group of random nucleotides to be flanked by fixed regions, optimizing spacing and possibilities for diverse secondary structures. The module-driven approach addresses the constraint of NGS read capacity (approximately $1 \times 10^9$ reads per lane) compared to the total number of possible sequences in the library ($4.29 \times 10^9$). By splitting the template into two modules (A and B) post-selection, each with eight random nucleotides (65,536 possible sequences per module), we ensure a statistically robust analysis using NGS. Prior to synthesis, the library design underwent computational validation using Vienna RNAfold to predict secondary structures of a subset of possible sequences, ensuring structural diversity (23) (Fig 2).

### Target preparation

**Interleukin 6 (IL6).** Targets for selection were immobilized on resin scaffold. Recombinant IL6 with an N-terminal his tag (Fitzgerald, Biosynth Ltd.) was immobilized on His-Pur™ Ni-NTA Resin (ThermoScientific) following a modified purification protocol.

IL6 (50 μg) was prepared in 200 μL of binding buffer (20 mM sodium phosphate, 300 mM sodium chloride, pH 7.4) and incubated with 70 μL of Ni-NTA Resin for 2 hr at room temperature with agitation. Unbound protein was collected by centrifuging at $700 \times g$ for 2 min and removing the supernatant. The resin was washed once with 140 μL of binding buffer. Remaining active sites were blocked with 2 mM imidazole, incubating for 1 hr at room temperature with rotation. After blocking resin was washed with 1x PBS and resuspended as a 50:50 slurry in 1x PBS.

**Human serum albumin (HSA).** HSA (Sigma-Aldrich Canada) was immobilized on Ultralink Biosupport (ThermoScientific), which uses azlactone functional groups to bind primary amines on proteins, following user guide recommendations. Solubilized HSA was prepared to a final concentration of 2 mg/mL in conjugation buffer (500 mM sodium citrate, 200 mM sodium bicarbonate, pH 8.5). 200 μL of protein solution was added to 8.8 mg of dry UltraLink Beads (swell volume of 8 μL/mg) and incubated for 2 hr at room temperature with agitation. Unbound protein was removed by centrifuging at $1200 \times g$ for 5 min. The resin was washed once with 140 μL of conjugation buffer. Remaining active sites were blocked overnight with 1 M Tris (which contains a primary amine) at 4 °C. After blocking the resin was washed and stored in PBS. This method results in multiple orientations of immobilized HSA, presenting various epitopes for aptamer binding.

## Neomer selection

Prior to selection, the Neomer library was denatured at 95 °C for 5 min, cooled on ice for 10 min, and equilibrated to room temperature. Selection was conducted in a 1 mL column fitted with a 20 μm frit. The refolded library (7.15 pmoles) was incubated with 10 μL of IL6-Ni-NTA resin (223 pmol IL6) in 100 μL of selection buffer (10 mM HEPES, 120 mM sodium chloride, 5 mM potassium chloride, 5 mM magnesium chloride, pH 7.6) for 30 min at room temperature with agitation. The unbound library was discarded in the flowthrough and the column washed three times with 500 μL of selection buffer. Bound library was eluted by adding 200 μL of 6 M urea to the column, heating at 85 °C for 5 min and collecting the flowthrough. The elution was repeated, and the eluents pooled together. The eluted library was purified using the GeneJet PCR Purification Kit (ThermoFisher) and eluted with 50 μL of MilliQ filtered water. The library was brought up to 100 μL in selection buffer and incubated with another 10 μL of IL6-Ni-NTA resin. The purified library was eluted in 400 μL of MilliQ filtered water and stored in a salinized vial. Selection against HSA was conducted using the same procedure with the following changes. The library was incubated with 10 μL of HSA-Ultralink Biosupport (369 pmol HSA) in 100 μL of PBS. The purified library was eluted in 400 μL of MilliQ filtered water. Each selection as described was conducted in triplicate for each target.

## NGS preparation

Nested PCR primers were applied to each selected DNA library for sequence identification. The sequences were amplified (S2 Table), isolated from a 20% acrylamide gel, and purified for sequencing. For purification of the DNA, NGS2 PCR products were run on 20% polyacrylamide gels at 150V for 5 hr. The target band was excised from the gel, fragmented, and stored in TE buffer in a salinized vial for 3 days to elute the DNA. The DNA was subsequently purified using a Genejet PCR purification kit. The DNA was purified following standard protocol and eluted from the column with 30 μl of water. 10 μl of the DNA is run on a 10% polyacrylamide gel to determine concentration of each library. Sequence analysis was completed by the Hospital for Sick Children (Toronto, CA) using Illumina HiSeq 2500.

Negative control selections of the library against UltraLink resin and nickel resin without immobilized proteins was also performed in triplicate as well as the unselected naïve library.

## Statistical analysis of selection

Next-generation sequencing of all libraries was performed by The Centre for Applied Genomics at The Hospital for Sick Children, Toronto, Canada. Sequencing data in FASTQ format was converted to FASTA, and a proprietary Python script was used to determine the copy number of each of the 65,536 sequences in modules A and B. The frequency of each module sequence was calculated by dividing the copy number by the total reads for that module (see Equation 1). The frequency of each of the $4.29 \times 10^9$ sequences in the original library was estimated using an outer product calculation of the module A and B frequencies.

$$\text{Frequency} = \frac{\text{Copy number}}{\text{Total read count}} \tag{1}$$

The average frequency of each of the $4.29 \times 10^9$ sequences was determined for all three target and naïve or counter-target samples in each selection, along with the standard deviation of the samples from each condition. The difference between the average frequency of each sequence in the target samples and the counter-target samples was calculated and divided by the average standard deviation of these frequencies to generate a z-score (see Equation 2). The top 10,000 sequences based on z-scores were then identified.

$$Z_{ij} = \frac{\overline{X}_{\text{Target}_{ij}} - \overline{X}_{\text{Naïve}_{ij}}}{\left[0.5 \times \left(s_{\text{Target}_{ij}} + s_{\text{Naïve}_{ij}}\right)\right]} \tag{2}$$

The performance of the top 10,000 sequences was evaluated against counter-targets (HSA, Ultralink resin, and nickel resin) using fold analysis. The average frequency of each sequence for each counter-target was divided by its average frequency for the positive target, yielding a fold value (see Equation 3).

$$\text{Fold} = \frac{\text{Frequency}_{\text{Target}}}{\text{Frequency}_{\text{Counter Target}}} \tag{3}$$

Where Target is the average frequency of each of the top 10,000 predicted aptamers for IL6, and Counter Target is the corresponding average frequency of the same sequence in each of full counter-target's full selection.

## Secondary structure analysis

1000 unique sequences were generated using both the Neomer template and a SELEX template derived from SomaScan, incorporating identical primer recognition regions and a 40-nucleotide random region [15]. Dotbracket structures for these were calculated using RNAfold 2.5.0 with parameters specifying a temperature of 22 °C and a DNA folding specific energy profile [22]. Secondary structure annotation for these sequences was computed using bpRNA under default parameters [23].

## Visualisation of IL6 aptamer analysis and secondary structure analysis

Data was processed in python 3.9.13 and loaded into R studio (version 2023.09.1 + 494) where they were analysed and visualised using ggplot and other packages [26–29].

## Aptamer library template visualisation

Secondary structure dotbracket information was computed using RNAfold v2.5.0 [22] with parameters specifying a temperature of 22 °C and a DNA folding specific energy profile and then inputted into FORNA [25]. A custom colour palette was chosen to highlight random regions in each library.

## General SPRI materials and methods

Binding assays were performed using the SPRi system (OpenPlex, HORIBA France). Bare gold biochips were purchased from Xantac Bioanalytics (Düsseldorf, Germany). All thiolated aptamers were synthesized and purchased from Integrated DNA Technology (IDT) (Lowa, USA). Recombinant Human IL-6 protein was purchased from Peprotech (Cranbury, USA) Albumin from human serum (HSA) was purchased from Sigma-Aldrich Canada. The 1x Running Buffer (pH5.7 ± 0.1 at 25 °C) contains 400 mM 6-aminohexanoic acid (EACA) (Sigma-Aldrich Canada), 2 mM Triethylenetetramine (TETA) (Sigma-Aldrich Canada) and 100 mM NaCl (Sigma-Aldrich Canada). The blocking agent 6-Mercapto-1-hexanol and Phosphate-Buffered Saline (PBS) (Sigma-Aldrich Canada) (10.14 mM $Na_2PO_4$, 137 mM NaCl, 2.68 mM KCl, pH 7.4) at 25 °C. The calibration solution in 1x running buffer contains 2mg/ mL Sucrose (Sigma-Aldrich Canada) and the regeneration solution for gold chip is 1% (w/v) Sodium Dodecyl Sulfate (SDS) (Sigma-Aldrich Canada).

## Preparation of sensor chip

The sensor chip and purified salt free thiolated aptamers were brought to room temperature prior to spotting, aptamers were immobilized on the gold surface in triplicate at 10 nL/ spot with a concentration of 5 μM. The gold chip was allowed a 1-hour incubation time for efficient immobilization at room temperature and at an approximate humidity of 80%. 1mM 6-mercapto-1-hexanol in 1x PBS was used to facilitate blocking of the functionalized surface of gold chip for an hour. The prepared sensor chip was then placed into the instrument.

## SPRi measurement

Surface plasmon resonance can be explained by an optical detection process that happens when a gold-layered prism is hit by a polarized light which facilitates the conversion from light photons into surface plasmon waves. This allows the measurement of a change of resonance condition due to any interaction between immobilized probes and targets. OpenPlex SPRi systems allows real-time quantification and monitoring of the biomolecular interactions. The method is to calculate the angle where the greatest variation of reflectivity Δ%R will be and adjusting this angle to monitor kinetics activity which is the amount of reflectivity versus time. A CCD camera is used in this system to visualize the spots and monitor their reflectivity simultaneously.

The biochip was loaded into the flow cell and target prepared in 1x running buffer was injected into the 200 μL sample loop at a flow rate of 50 μL/min. One completed run is 9 minutes, this means that the target was in flow over the aptamers for 240 seconds which is considered as association phase (ka). The remaining 300 seconds which is disassociation phase (kd) indicates the target was no longer in flow. Equilibrium dissociation constant (kD) was calculated using the equation kD = kd/ka. SPRi response recorded was the average response of 3 spots of each immobilized aptamer, the non-specific (negative control) signal was subtracted during data analysis. A calibration coefficient was obtained by injecting 2 mg/ mL sucrose in running buffer to collect a variation of the reflectivity of individual spot which

used to fix all spots to the same change in reflectivity, this promoted a kinetics camera angle of 58.16 degree. 1% (w/v) SDS solution was utilized to regenerate the functionalized surface, removal efficiency was measured by SPR signals, once the signal was restored to baseline the sensor chip was ready for another target injection. All injections were conducted at 25 °C (room temperature).

## Supporting information

**S1 Fig. The probability that any given nucleotide would be contained within a specific structural motif was similar between SELEX and Neomer library designs.** A) Average secondary structure motif frequency in each sequence across 1000 randomly generated SELEX sequences. B) Average contiguous nucleotide length of secondary structure motifs in each sequence across 1000 randomly generated SELEX sequences. C) Average secondary structure motif frequency in each sequence across 1000 randomly generated Neomer sequences. D) Average contiguous nucleotide length of secondary structure motifs in each sequence across 1000 randomly generated Neomer sequences.
(TIF)

**S2 Fig. Average Shannon diversity index value across position for the SELEX and Neomer template.** The template per position is plotted out below the plot.
(TIF)

**S1 Table. Reads from NGS.**
(DOCX)

**S2 Table. Primers used in NGS preparation.**
(DOCX)

## Author contributions

**Conceptualization:** Gregory Penner.

**Data curation:** Cathal Meehan, Erika L. Hamilton, Soizic Lecocq, Yisi An.

**Formal analysis:** Gregory Penner, Cathal Meehan, Chloé G. Mansour, Cole J. Drake, Yisi An.

**Investigation:** Gregory Penner, Cathal Meehan.

**Methodology:** Erika L. Hamilton, Soizic Lecocq.

**Software:** Cathal Meehan, Chloé G. Mansour.

**Supervision:** Gregory Penner.

**Visualization:** Cathal Meehan, Chloé G. Mansour.

**Writing – original draft:** Gregory Penner, Cathal Meehan, Chloé G. Mansour, Emily Rodrigues.

**Writing – review & editing:** Gregory Penner, Cathal Meehan, Chloé G. Mansour.

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
