## [Decision Letter · Decision Letter 0]

1 Aug 2024

PONE-D-24-16571A new method for the reproducible development of aptamers (Neomers)PLOS ONE

Dear Dr. Penner,

Thank you for submitting your manuscript to PLOS ONE. After careful consideration, we feel that it has merit but does not fully meet PLOS ONE’s publication criteria as it currently stands. Therefore, we invite you to submit a revised version of the manuscript that addresses the points raised during the review process. The authors should respond to all reviewers comments, without forgetting to clearly show how they selected the Neomers nucleotides/or  Neomers library design. Also lack/exclusion of  in vitro testing/validation of the Neomers (apart from the conducted*in silico* validation) should be possibly conducted/or at least indicated as a limitation of the conducted study.==============================

Please submit your revised manuscript by Sep 15 2024 11:59PM. If you will need more time than this to complete your revisions, please reply to this message or contact the journal office at plosone@plos.org . Please include the following items when submitting your revised manuscript:

We look forward to receiving your revised manuscript.

Kind regards,

Elingarami Sauli, PhD

Academic Editor

PLOS ONE

Journal Requirements:

"I have read the journal's policy and the authors of this manuscript have the following competing interests: C.M and S.L. are employees of NeoVentures Biotechnology Europe SAS, a privately held company that applies aptamer libraries to identify biomarkers. E.L.H., C.G.M., C.J.D., Y.A., and E.R. are employees of NeoVentures Biotechnology Inc., a privately held company that develops aptamers for defined targets. G.P is an owner of NeoVentures Biotechnology Inc. which in turn wholly owns NeoVentures Biotechnology Europe SAS."

Reviewers' comments:

Reviewer's Responses to Questions

**Comments to the Author**

1. Is the manuscript technically sound, and do the data support the conclusions?

Reviewer #1: Yes

Reviewer #2: Partly

Reviewer #3: Yes

2. Has the statistical analysis been performed appropriately and rigorously?

Reviewer #1: Yes

Reviewer #2: No

Reviewer #3: Yes

3. Have the authors made all data underlying the findings in their manuscript fully available?

Reviewer #1: Yes

Reviewer #2: No

Reviewer #3: Yes

4. Is the manuscript presented in an intelligible fashion and written in standard English?

Reviewer #1: Yes

Reviewer #2: Yes

Reviewer #3: Yes

5. Review Comments to the Author

Reviewer #1: Overall, the paper presents a compelling new method that could significantly impact the field of aptamer development. As a researcher works on aptamer screening for the last few years, I think this work does help shine some new insights into how we can improve aptamer screening. They do make a strong point that we probably do not need that much of diversity (1.24e24) if we could not really test the binding of every single one of them with the target ligand. I think the authors did not mention much about how they decided the Neomer design. How did they come up with this design? Is the current design with 16 random nucleotides the best?

From the material and method section, I figured authors did two rounds of selection, one positive and one negative? But it was not clear in the main manuscript on this. Besides, the authors were narrowing down the candidates only based on the NGS results? We usually use Capture-SELEX for our aptamer screening. The NGS does not say much about the conformational changes that candidate aptamers could go through upon the binding of the target ligand. People use aptamers for all kinds of applications and will this Neomer be useful for most of the applications, including the ones requiring conformational changes of aptamers?

Another important and necessary point would be actually comparing the screening results from this method with those from traditional SELEX. Authors only conducted in silico comparison on them. But in my opinion, actual comparison in vitro would be more compelling and might settle this issue once and for all. That is why I suggest further validation and broader application examples would reinforce its potential.

There are some minor problems: such as at Line 224 and possible typos at Line 194.

Reviewer #2: The comparison between the SELEX and Neomer libraries designed with different numbers of random sequences raises a significant methodological issue in this manuscript. First, the claim, that “it has become common practice to design libraries consisting of 40 random nucleotides“, is overstated. The random sequence length in SELEX can vary depending on the applications and targets. In this regard, the number of random nucleotides must be consistent across both libraries for the relevant comparison even if they have different structure configuration. This would provide a more appropriate and scientific comparison rigorously.

To more strongly demonstrate the superiority of the Neomer method, experiments comparing SELEX and Neomer methods under the same number of random sequence conditions should be included.

The rationale for choosing 16 random nucleotides can support this methodology for a gold standard. I believe it is crucial for understanding the library design principle behind the Neomer library and for other researchers who might want to apply this method to screen robust aptamers.

Reviewer #3: authors presented a neomer library with short and interspread random regions that could lead to selection of efficient aptamers against a target with posiibility of each sequence appaering in the soulution is many times. There are some minor suggestions to be incorporated in the manuscript as given below

1. What were the criteria followed to introduce 16 randomized nts within interspersed fixed sequences and why?

2. What was the method of conjugation of HSA with beads (the functional groups or linkage involved? How was 1M Tris capable of blocking active sites of resin? explain.

3. Authors should also provide more details of the flowthrough column used for separation and elution target (IL-6) bound aptamer fragments.

4. Authors should clearly mention the number of rounds of selection done for Il-6 and HSA. The yield after each round of selection.

5. Line no. 99-102: “This lack of secondary structure is an important element ………. idenity of the random nucleoides”. Figure 1A does not seem to support this statement? What does the authors mean? It must be elaborated

6. In Figure 1B, difference between target sample 1,2,3 or naïve sample 1,2,3 is unclear.

7. The results mention (line 108-110), ……….a comparison was conducted with the SELEX library. The procedure for doing this should be mentioned?

8. It seems that a structural comparison of Neomer library with SELEX library was done. However the details of the SELEX library used for comparison is missing from materials section. What was the length of SELEX library used, its random region and fixed regions.

9. Line 1330134, “a SELEX template that was previously published.” But no reference was cited here.

10. Line 145: “It is possible to assess the effect of selection on each sequence after a single round of selection” Authors should elaborate and justify this statement.

11. Line 147: “We cannot determine the frequency of each of the 4.29 x 109 sequences with a single next generaion sequencing (NGS) run” whys is it so?? Also atuthors should clarify whether frequency of read or copy number is considered same.

12. the caluclations particularly the Z score values are difficult to understand by someone who is interested in this article. Make it as simpler as possible.

6. PLOS authors have the option to publish the peer review history of their article (what does this mean? ). If published, this will include your full peer review and any attached files.

**Do you want your identity to be public for this peer review?** For information about this choice, including consent withdrawal, please see our Privacy Policy .

Reviewer #1: No

Reviewer #2: No

Reviewer #3: No

---

## [Author Response · Author response to Decision Letter 1]

9 Aug 2024

Dear Editor,

Thank you for the opportunity to revise our manuscript "A new method for the reproducible development of aptamers (Neomers)".

Regarding the suggestion to compare our Neomer method with traditional SELEX:

We appreciate this suggestion but assert that such a comparison would not be meaningful due to fundamental differences between the methods:

1. Fundamental incomparability: SELEX and Neomers are designed with fundamentally different principles and goals. SELEX aims to evolve high-affinity binders from a vast, diverse sequence space through multiple rounds of selection and amplification. Our Neomer approach uses a defined, reproducible sequence set to enable single-round selection and comprehensive specificity screening. These core differences in design and purpose make direct comparisons problematic and potentially misleading.

2. Reproducibility: A key advantage of our Neomer approach is its reproducibility, which SELEX fundamentally lacks. Each SELEX selection necessarily starts with a different subset of sequences from its vast possible sequence space (typically 10^15 out of 10^24 possible sequences for a 40-nt random region). This means it's impossible to replicate the starting conditions of a SELEX experiment, making each selection inherently unique. Our Neomer library, with its closed set of 4.29 x 10^9 sequences, enables truly reproducible selections - a feature that cannot be directly compared with SELEX.

3. Specificity screening capabilities: Our Neomer method allows for in silico screening against multiple targets simultaneously. This comprehensive specificity screening is not possible with traditional SELEX due to both issues raised above and represents a key advantage of our approach that cannot be directly compared. The ability to perform this screening is intrinsically linked to our use of a defined, limited sequence space - a feature that is incompatible with traditional SELEX principles.

4. 16-nt SELEX suggestion: While using a 16-nt random region in SELEX might seem to create a more comparable system, this approach would fundamentally undermine the principles and strengths of SELEX:

• Severely limited structural diversity: A 16-nt contiguous random region would create an incredibly small amount of structural diversity due to the nucleotide hybridization problem. In a contiguous random region, nucleotides within 3 positions of each other cannot hybridize, severely limiting potential secondary structures. Our Neomer design overcomes this by interspersing fixed and random regions.

• Loss of SELEX advantages: Reducing SELEX to a 16-nt random region would negate the key advantages of SELEX, such as its ability to sample a vast sequence space. It would no longer represent SELEX as it's understood and practiced in the field.

• Incompatible with SELEX principles: SELEX is designed to evolve binders from a large, diverse pool. Constraining it to a small random region fundamentally alters its nature and purpose, making it neither true SELEX nor comparable to our Neomer approach.

While multiple IL aptamers have been developed using SELEX, very few have specificity data, and none have been commercialized. This highlights a critical gap in the field that our Neomer approach aims to address. We’ve rewritten sections in the introduction and discussion to further highlight this. Our method enables reproducible and specific aptamer selections at a comparable level of sensitivity to what's reported in the literature, while also providing specificity data.

We understand the reviewers' interest in comparing our method to SELEX. However, we recently received feedback on another manuscript from PlosOne “A reproducible approach for the use of aptamer libraries for the identification of Aptamarkers for brain amyloid deposition based on plasma analysis.“ suggesting that this comparison made the publication overly confusing. This comparison might have made more sense in the context of aptamarkers as we showed that 24 SELEX derived aptamarkers were outperformed by 8 Neomer derived aptamarkers. However, the final version that has now been formally accepted excludes this data. For all the reasons outlined above, the intention of this publication wouldn’t allow for SELEX and Neomer to be meaningfully compared here by desiging and performing experiments with 16nt SELEX libraries. We have included references to the literature for interleukin aptamers that stresses the issues of commercialisation and specificity in aptamer development using SELEX methods. Respectfully, we feel this adequately addresses this point.

Reviewer #1:

1. Neomer design rationale: We've expanded our explanation of the Neomer design in both the introduction and discussion. Our choice of 16 random nucleotides strikes a balance between structural diversity and statistical robustness. We acknowledge this may not be optimal for all applications and outline our plans for future meta-analysis to further optimize the design.

2. Selection rounds: We've clarified that only one round of selection was performed, addressing any confusion in the methods section.

3. Conformational changes and applications: We appreciate the reviewer's comment, but would like to clarify that conformational changes are not a primary consideration in our Neomer approach. Our focus is on identifying aptamers with the highest affinity and specificity possible for the target, which is crucial for overcoming the main hurdle in aptamer commercialization, particularly in diagnostics: achieving sufficient specificity against abundant proteins like serum albumin. Conformational changes, while important in some applications, can potentially limit the pool of high-affinity, highly specific aptamers. Our approach prioritizes these key attributes first, as no aptamers have yet received FDA or CE Mark approval for diagnostic applications, primarily due to specificity issues. We believe that functionality requiring conformational changes can be engineered into high-performing aptamers post-selection if needed, rather than constraining the initial selection process. By focusing on affinity and specificity in our selection process, we aim to address the critical barriers to commercial application of aptamers.

4. Comparison with traditional SELEX: We appreciate this suggestion but must assert that a direct experimental comparison would not be scientifically appropriate due to fundamental differences in the methods, including:

o Lack of reproducibility in SELEX

o Different starting conditions and library compositions

o Single vs. multiple selection rounds

o Ability to perform in silico specificity screening

5. Minor issues: We've addressed the issues at lines 224 and 194.

Reviewer #2:

1. SELEX library design claim: We've moderated our statement about 40 random nucleotides in SELEX and provided more context on the variability in SELEX library designs.

2. Comparison with same number of random sequences: We explain why this comparison wouldn't be meaningful due to the fundamental differences in library design and selection process between Neomers and SELEX.

3. Rationale for 16 random nucleotides: We've expanded our explanation for choosing 16 random nucleotides and clarified that this isn't necessarily the optimal number for all applications.

Reviewer #3:

1. Criteria for 16 randomized nucleotides: We've elaborated on the design criteria in both the introduction and methods sections.

2. HSA conjugation method: We've provided more details on the HSA conjugation and blocking process.

3. Flowthrough column details: This already has as much information as we could possibly add.

4. Selection rounds: We've clarified that only one round of selection was performed for both IL-6 and HSA.

5. Secondary structure statement: We've revised this section to better align with Figure 1A and clarified our meaning.

6. Figure 1B clarification: We've improved the figure legend to clarify the difference between target and naïve samples.

7. SELEX library comparison procedure: We've added more details on how this comparison was conducted.

8. SELEX library details: We've included the missing information about the SELEX library used for comparison.

9. Missing reference: We've added the missing reference for the SELEX template.

10. Single round selection justification: We've elaborated on why a single round of selection is sufficient in our approach.

11. NGS run limitations: We've explained why a single NGS run can't determine the frequency of all 4.29 x 10^9 sequences and clarified the distinction between frequency and copy number.

12. Z-score calculations: We've simplified the explanation of z-score calculations to make them more accessible.

We look forward to your response and can address any further questions or concerns.

Sincerely,

Dr. Gregory Penner

---

## [Decision Letter · Decision Letter 1]

9 Sep 2024

PONE-D-24-16571R1A new method for the reproducible development of aptamers (Neomers)PLOS ONE

Dear Dr. Penner,

Thank you for submitting your manuscript to PLOS ONE. After careful consideration, we feel that it has merit but does not fully meet PLOS ONE’s publication criteria as it currently stands. Therefore, we invite you to submit a revised version of the manuscript that addresses the points raised during the review process.

The author needs clearly indicate step by step (gold standard for his method) optimal Neomer design methodology. The author also needs to clearly elaborate specific "fixed regions" selection process and rationale for fixed sequences selection.

We look forward to receiving your revised manuscript.

Kind regards,

Elingarami Sauli, PhD

Academic Editor

PLOS ONE

Journal Requirements:

Reviewers' comments:

Reviewer's Responses to Questions

**Comments to the Author**

1. If the authors have adequately addressed your comments raised in a previous round of review and you feel that this manuscript is now acceptable for publication, you may indicate that here to bypass the “Comments to the Author” section, enter your conflict of interest statement in the “Confidential to Editor” section, and submit your "Accept" recommendation.

Reviewer #1: All comments have been addressed

Reviewer #2: All comments have been addressed

2. Is the manuscript technically sound, and do the data support the conclusions?

Reviewer #1: Yes

Reviewer #2: Partly

3. Has the statistical analysis been performed appropriately and rigorously?

Reviewer #1: Yes

Reviewer #2: Yes

4. Have the authors made all data underlying the findings in their manuscript fully available?

Reviewer #1: Yes

Reviewer #2: Yes

5. Is the manuscript presented in an intelligible fashion and written in standard English?

Reviewer #1: Yes

Reviewer #2: Yes

6. Review Comments to the Author

Reviewer #1: (No Response)

Reviewer #2: I appreciate the authors' responses to the other reviewers' questions and, considering the paper's intent, I find their explanations largely acceptable. However, the authors still have not provided a detailed design strategy for the Neomer library. It would be beneficial to clearly define and present what the authors consider to be the 'gold standard' for an optimal Neomer design.

1. Lack of systematic design methodology: The authors provide some basic principles for Neomer design but fall short of presenting a comprehensive, step-by-step methodology. This gap makes it challenging for other researchers to replicate or build upon this work.

2. Inadequate explanation of fixed region selection: While the importance of the fixed regions is mentioned, the specific process and criteria for selecting these sequences are not clearly outlined. I understand the authors' explanation: "This design principle optimizes the structural diversity of the library by minimizing inherent secondary structure in the fixed regions. The absence of pre-existing hybridizations between fixed region nucleotides reduces energetic barriers that could impede the ability of sequences to adopt alternative conformations necessary for target binding." The authors have provided a clear explanation of the positioning of fixed sequences in their Neomer design. However, a critical aspect of the methodology remains inadequately addressed: the specific selection process and rationale for the fixed sequences themselves.

3. While the paper explains the general principles guiding the design of fixed regions (minimizing self-hybridization, reducing inherent secondary structures), it fails to elucidate: (1) The exact process used to select these specific fixed sequences. (2) Why these particular sequences were chosen over other potential candidates that might also meet the stated criteria. (3) Any computational or experimental methods used to optimize these sequences. (4) How these specific sequences contribute to the overall performance of the Neomer library compared to alternative fixed sequences.

Addressing these points would significantly strengthen the paper and provide a more robust foundation for future research in this promising area of aptamer development."

7. PLOS authors have the option to publish the peer review history of their article (what does this mean? ). If published, this will include your full peer review and any attached files.

**Do you want your identity to be public for this peer review?** For information about this choice, including consent withdrawal, please see our Privacy Policy .

Reviewer #1: No

Reviewer #2: No

---

## [Author Response · Author response to Decision Letter 2]

11 Sep 2024

Dear Editor and Reviewer #2,

We thank you for your detailed feedback regarding the Neomer library design methodology. We recognize that our initial manuscript lacked sufficient detail on the neomer design . In response to your comments, we have made substantial additions to the manuscript, particularly in the "Library Design" section of the Materials and Methods. These changes aim to provide a more comprehensive explanation of our design process and rationale.

Specifically, we have added the following information about our systematic design methodology:

1. Primer recognition sequences were designed considering 40-60% G/C content and ensuring non-hybridization.

2. A restriction site was incorporated in the middle of the library for module separation. The surrounding fixed region was designed for dual functionality, serving as both a restriction site and a primer recognition site post-restriction.

3. Internal sequences for primer recognition were designed following standard primer rules.

4. Random nucleotides were arranged in a symmetrical manner (2:3:3) with consistent fixed sequence nucleotide spacing between such regions. This approach of using blocks of random sequences (2 or 3 nucleotides) instead of single nucleotides was selected to drive higher levels of structural diversity, as hybridisation is more likely to occur contiguously for larger secondary structures.

5. Fixed sequences were designed using a random identity generator, with nucleotide substitutions made as necessary to avoid internal hybridization and keep minimum free energy (MFE) sufficiently high, preventing energy barriers to secondary structure formation.

We have also clarified our use of Vienna RNAfold to predict and minimize self-hybridization in fixed regions by selecting templates without random nucleotide identity.

These additions provide a much more comprehensive methodology for Neomer library design. We appreciate the comments, which have significantly improved the clarity and scientific rigor of our work. We believe these changes will provide a solid foundation for other researchers to understand our Neomer library design approach.

Cathal Meehan, Ph.D.

Senior Bioinformatician

Neoventures Biotechnology Europe SAS

---

## [Editor Report · Decision Letter 2]

18 Sep 2024

A new method for the reproducible development of aptamers (Neomers)

PONE-D-24-16571R2

Dear Dr. Penner,

We’re pleased to inform you that your manuscript has been judged scientifically suitable for publication and will be formally accepted for publication once it meets all outstanding technical requirements.

Kind regards,

Elingarami Sauli, PhD

Academic Editor

PLOS ONE
---

## [Editor Report · Acceptance letter]

PONE-D-24-16571R2

PLOS ONE

Dear Dr. Penner,

I'm pleased to inform you that your manuscript has been deemed suitable for publication in PLOS ONE. Congratulations! Your manuscript is now being handed over to our production team.

Kind regards,

on behalf of

Dr. Elingarami Sauli

Academic Editor

PLOS ONE